# The retinal ipRGC-preoptic circuit mediates the acute effect of light on sleep

Ze Zhang [1,2]✉, Corinne Beier [1,2], Tenley Weil [1] & Samer Hattar [1]✉

Light regulates daily sleep rhythms by a neural circuit that connects intrinsically photo-sensitive retinal ganglion cells (ipRGCs) to the circadian pacemaker, the suprachiasmatic nucleus. Light, however, also acutely affects sleep in a circadian-independent manner. The neural circuits involving the acute effect of light on sleep remain unknown. Here we uncovered a neural circuit that drives this acute light response, independent of the supra-chiasmatic nucleus, but still through ipRGCs. We show that ipRGCs substantially innervate the preoptic area (POA) to mediate the acute light effect on sleep in mice. Consistently, activation of either the POA projecting ipRGCs or the light-responsive POA neurons increased non-rapid eye movement (NREM) sleep without influencing REM sleep. In addition, inhibition of the light-responsive POA neurons blocked the acute light effects on NREM sleep. The predominant light-responsive POA neurons that receive ipRGC input belong to the corticotropin-releasing hormone subpopulation. Remarkably, the light-responsive POA neurons are inhibitory and project to well-known wakefulness-promoting brain regions, such as the tuberomammillary nucleus and the lateral hypothalamus. Therefore, activation of the ipRGC-POA circuit inhibits arousal brain regions to drive light-induced NREM sleep. Our findings reveal a functional retina-brain circuit that is both necessary and sufficient for the acute effect of light on sleep.

[1] Section on Light and Circadian Rhythms, National Institute of Mental Health (NIMH), National Institutes of Health (NIH), Bethesda, MD, USA. [2]These authors contributed equally: Ze Zhang, Corinne Beier. ✉email: ze.zhang@nih.gov; samer.hattar@nih.gov

Nighttime light exposure has become widespread in industrialized societies[1,2]. Artificial light at night profoundly disrupts sleep in humans[3–6]. Melanopsin (OPN4)-expressing intrinsically photosensitive retinal ganglion cells (ipRGCs) convey retinal photoreceptor-driven light input to the brain to drive the photoentrainment of sleep rhythms and also to drive the acute effect of light on sleep[7–16]. The circuit that drives the photoentrainment of sleep rhythms is well defined and it is known to involve the suprachiasmatic nucleus (SCN), the circadian pacemaker[8,17]. However, our recent study showed that the SCN is not sufficient for acute effect of light on sleep[18]. Thus, the neural circuits that underlie the acute effect of light on sleep remain to be understood. The preoptic area (POA) has been known to be critical for sleep regulation[19–26] and is the predominant sleep center[27]. Damage to the POA was found to be related to insomnia in human patients[28]. However, the POA is thought to be very sparsely innervated by the retina based on previous studies using traditional tracing methods[29,30], which undermined the role of the POA in the acute light effects on sleep. Therefore, no studies thus far have assessed whether the POA mediates the acute effect of light on sleep[31]. Here, using a high-efficiency tracing strategy, we unexpectedly find that ipRGCs have a much more substantial innervation to the POA than originally reported. Importantly, ipRGCs only target a subpopulation of POA neurons that influence sleep. Activation of the ipRGCs that project to the POA or activation of the light-responsive POA neurons themselves both increase the amount of non-rapid eye movement (NREM) sleep. Additionally, inhibition of the light-responsive POA neurons blocks the acute light effects on NREM sleep. Thus, our study reveals that a neural circuit from ipRGCs to the POA is predominantly responsible for the acute effect of light on NREM sleep.

## Results

**Activation of ipRGCs that project to the POA increases NREM sleep.** First, we assessed the ipRGC projections to the POA by intravitreally injecting Cre-inducible adeno-associated virus (AAV) expressing the genetically encoded green fluorescent protein (AAV2-CAG-DIO-GFP) in $Opn4^{Cre}$ mice (Fig. 1a, b). We found that the MPO (medial preoptic area), the LPO (lateral preoptic area), and the VLPO (ventrolateral preoptic nucleus) received innervation from ipRGC fibers (Fig. 1c, d; Supplementary Fig. 1a, b). To assess the specificity of the viral manipulations, we intravitreally injected AAV2-CAG-DIO-GFP virus in $Opn4^{Cre}$ and CAG-FLEX-tdTomato (Ai9) mice ($Opn4^{Cre};Ai9$). We found that more than 95% (95.35% ± 2.75%) of the GFP antibody-positive cells are also OPN4 mCherry-expressing cells (Supplementary Fig. 1c–e). In addition, we also intravitreally injected the AAV2-CAG-DIO-GFP virus in wild-type (WT) mice and we found no GFP expression in the retina (Supplementary Fig. 2a), indicating that the virus is only expressed in cells that express the Cre recombinase. We next examined whether this ipRGC-POA circuit affects sleep. To specifically manipulate the ipRGC-POA circuit, we implemented a dual-injection strategy in $Opn4^{Cre}$ mice by injecting an AAV that expresses Flp recombinase in a Cre-dependent manner (AAV2-Ef1α-DIO-Flp) in the eye, and injecting a Flp-dependent retrograde AAV encoding engineered Gq-coupled hM3D receptor and enhanced GFP (AAVretro-Ef1α-fDIO-hM3D(Gq)-EGFP) in the POA (Fig. 1f, g). Importantly, all the cells that express EGFP from the retrograde POA injections are also OPN4 positive (Supplementary Fig. 1f–h), showing high specificity for our manipulations. To determine the injection site in the brain, a non-specific AAV expressing red fluorescent protein mCherry (AAV8-hSyn-mCherry) was mixed with AAVretro-Ef1α-fDIO-hM3D(Gq)-EGFP (Fig. 1f, h). This strategy allowed us to label ipRGCs that only project to the POA, as no Cherry expression was found in other downstream brain regions of ipRGCs, even ones that are in close proximity to the POA, including the SCN (Supplementary Fig. 3a, b). Although ipRGCs are diverse, we found that ipRGCs that project to the POA belong to the M1 subtype (Supplementary Fig. 1i), which is well characterized[8]. As a control, we show that the same viral injection strategy produced no GFP expression in the retina of WT mice that lack Cre expression in ipRGCs (Supplementary Fig. 2b). Therefore, the specific labeling of ipRGCs that project to the POA allowed us to use chemogenetics to test their role in sleep regulation. We measured sleep states on the basis of electro-encephalogram (EEG) and electromyogram (EMG) using a wireless recording system (Fig. 1e). The EEG during NREM sleep is dominated by high-amplitude, low-frequency activity and EMG shows low muscle tone. The REM sleep, also known as paradoxical sleep, features EEG that is similar to wakefulness with high-frequency, low-amplitude activity but EMG shows a complete muscle atonia. At zeitgeber time (ZT) 14, clozapine-N-oxide (CNO) injection induced a significant increase in NREM sleep relative to normal saline (NS) control (Fig. 1i–k), without influencing REM sleep (Fig. 1l, m). As a control, we injected a virus that does not contain DREADDs in $Opn4^{Cre}$ mice and found that CNO injection did not affect NREM or REM sleep (Supplementary Fig. 4). The average hourly increase in NREM sleep in Fig. 1k is similar in magnitude to that induced by light in previous publications[15,16], indicating a predominant role for ipRGCs that project to the POA in regulating NREM sleep in response to light.

**Stimulation of light-responsive POA neurons induces NREM sleep.** To further understand the ipRGC-POA circuit, we used a second complementary approach to that employed in Fig. 1, where we combined chemogenetics with Fos-2A-iCreER mice[32]. This approach leads to the chemogenetic activation of light-responsive POA neurons themselves instead of ipRGCs. We injected Cre-dependent AAV8-hSyn-DIO-hM3D(Gq)-mCherry in the POA of Fos-2A-iCreER mice. One week later, animals received a light pulse from ZT 14-24 and an injection of 4-Hydroxytamoxifen (4-OHT) at ZT 16 to allow the 4-OHT dependent recombinase CreER expressed from the Fos locus to be translocated to the nucleus and induce the expression of hM3D(Gq) and mCherry only in light-responsive neurons (Fig. 2a, b). Here, we adopted a 10-h light pulse during ZT 14-24 to maintain the animals in sleep state because the effect of 4-OHT can last for 6-12 hours[32] and the POA may also contain wakefulness related neurons based on a recent study[33]. It is important to note that this 10-hour light pulse is used only once, and animals recovered their circadian photoentrainment for 4 weeks before measuring sleep through EEG and EMG recording in response to chemogenetic activation. Injection of CNO at ZT 14 significantly increased NREM sleep compared to NS control (Fig. 2c–e), whereas REM sleep was not altered (Fig. 2f, g). To determine if NREM increases are represented in specific frequency bands, we analyzed the power density of NREM sleep in our experiments. We found that the power density was not significantly affected in control versus CNO groups (Supplementary Fig. 5). Similar to what we found in Fig. 1, the increase of NREM sleep is similar in magnitude to those induced by light in previous publications[15,16]. This effect was specific to the POA as no virus-mediated expression was found in other downstream brain regions of ipRGCs that are close to the POA (Supplementary Fig. 6a, b), confirming the role of the ipRGC-POA circuit in influencing NREM sleep.

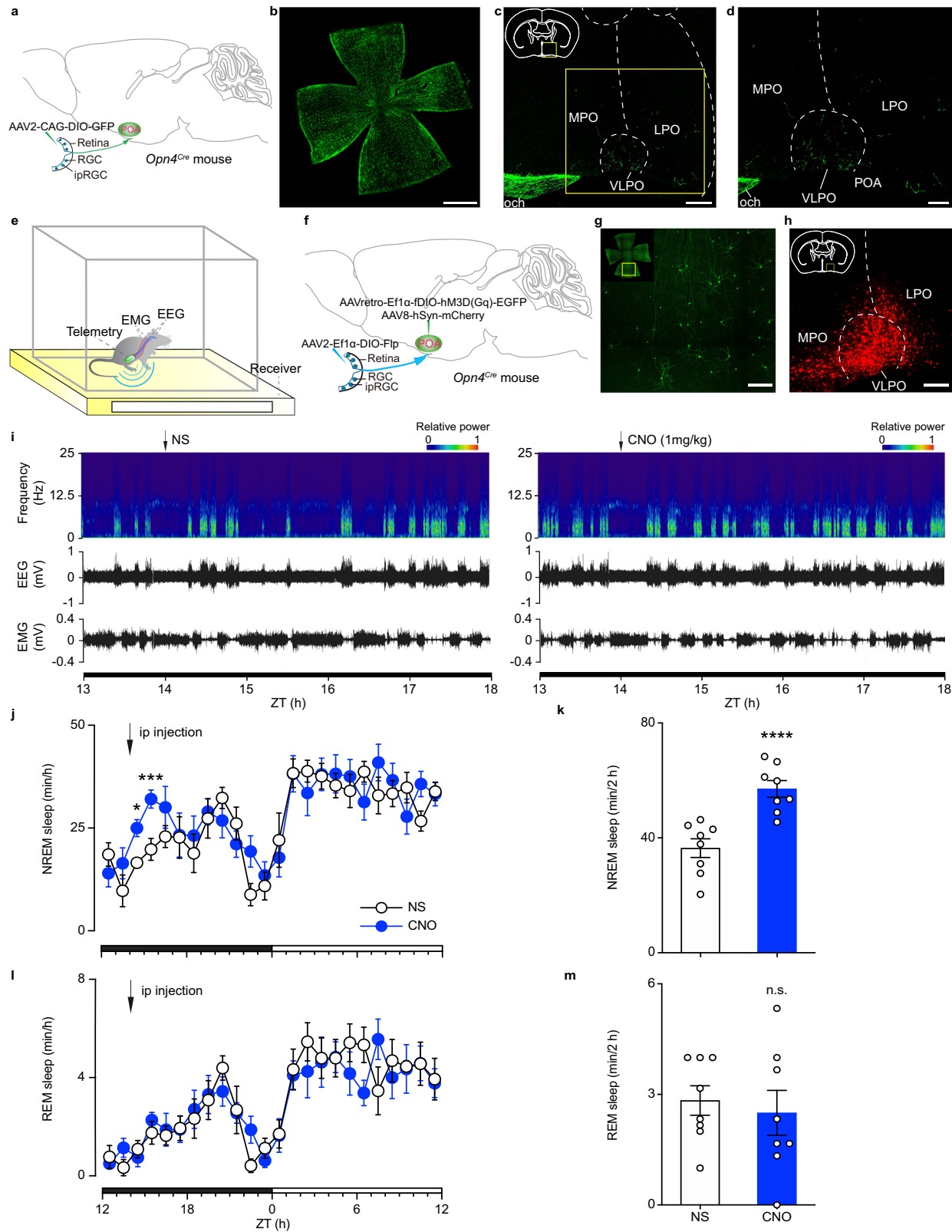

Although our results from Figs. 1 and 2 are conclusive, there is still a small possibility that the same POA neurons that are labeled by light pulse are also active in the absence of light, in the baseline dark condition, and hence can induce NREM sleep without the light pulse. To completely rule out this possibility, we performed the same experiment but simply eliminated the 10-hour light pulse. Specifically, Fos-2A-iCreER mice were injected

with AAV8-hSyn-DIO-hM3D(Gq)-mCherry and 4-OHT but without receiving a light pulse from ZT 14-24 (Supplementary Fig. 7a). Since mice sleep for less time at night, a few neurons in the POA were labeled (Supplementary Fig. 7b), which indicates that some of the POA neurons are active at night. However, in these mice, CNO injection had no effect on NREM or REM sleep as compared to NS control (Supplementary Fig. 7c, d), indicating

**Fig. 1 Activation of ipRGC-POA circuit induced NREM sleep but not REM sleep. a** Schematic of intravitreal virus injection. **b**, **c** Immunostaining of GFP in the retina ($n = 10$ retinas from 5 animals; Scale bar, 1000 µm) (**b**), the MPO, the LPO, and the VLPO ($n = 5$; scale bar, 200 µm) (**c**). The boxed region in **c** is enlarged in **d** ($n = 10$ retinas from 5 animals; scale bar, 100 µm). **e** Schematic showing electroencephalogram (EEG)/ electromyogram (EMG) wireless recording. **f** Schematic of dual injection strategy showing intravitreal and brain virus injections. **g** Immunostaining of EGFP in the retina ($n = 16$ retinas from 8 animals; scale bar, 200 µm). **h** Immunostaining of mCherry in the POA ($n = 8$ animals; scale bar, 200 µm). **i** Typical examples of relative EEG power (normalized by the highest power in each frequency band) and EEG/EMG traces over 4 h following normal saline (NS) injection (left) or clozapine-N-oxide (CNO) injection (right) at zeitgeber time (ZT) 14. **j** Time course changes in NREM sleep after NS or CNO injection ($n = 8$ animals; two-way ANOVA (ZT 14 to ZT 16), $F_{1, 14} = 22.45$, $P = 0.0003$, Bonferroni post hoc test, $*P < 0.05$, $***P < 0.001$). Each cycle represents the hourly mean ± SEM of sleep. The black and white bars on the x axes indicate light-off and light-on periods, respectively. **k**, Total time sleep in NREM sleep for 2 h after NS or CNO injection ($n = 8$ animals; two-tailed Student's t-test, $t_7 = 8.379$, $P = 0.000068$, $****P < 0.0001$). **l** Time course changes in REM sleep after NS or CNO injection ($n = 8$ animals; two-way ANOVA, $F_{1, 14} = 0.0821$, $P = 0.7786$, Bonferroni post hoc test). **m** Total time sleep in REM sleep for 2 h after NS or CNO injection ($n = 8$ animals; two-tailed Student's t test, $t_7 = 0.8913$, $P = 0.4024$). All error bars denote SEM. ipRGC, intrinsically photosensitive retinal ganglion cell; MPO, medial preoptic area; LPO, lateral preoptic area; VLPO, ventrolateral preoptic nucleus; och, optic chiasm; n.s., non-significant; NREM, non-rapid eye movement.

that the light pulse is required for the labeling of POA neurons that affect NREM sleep. Our two approaches determine that the ipRGC-POA circuit drives the acute effect of light on NREM sleep.

**Light-responsive POA neurons are necessary for light-induced NREM and influence daytime NREM sleep.** To examine whether the light-responsive POA neurons are necessary for light-induced sleep and for daytime sleep, we injected Cre-dependent AAV8-hSyn-DIO-hM4D(Gi)-mCherry in the POA of Fos-2A-iCreER mice. One week later, animals received a light pulse from ZT 14-24 and an injection of 4-OHT at ZT 16 to induce the expression of hM4D(Gi) and mCherry only in light-responsive POA neurons (Fig. 3a, b). Animals recovered their circadian photoentrainment for 4 weeks before measuring sleep through EEG and EMG recording in response to chemogenetic inhibition. A light pulse from ZT 14 to ZT 17 was given with either injected NS or CNO at ZT 14. In NS injection group, both NREM and REM sleep were significantly increased compared to dark control (Fig. 3c–f). This is consistent with previous reports that acute light pulse at night can increase both NREM and REM sleep[12,16]. However, in CNO injected group, the effect of acute light on NREM sleep was abolished, whereas the acute light effect on REM sleep was maintained (Fig. 3c–f), suggesting that inhibition of light-responsive POA neurons can abolish light-induced NREM sleep without affecting REM sleep. This effect was specific to the POA as no virus-mediated expression was found in other downstream brain regions of ipRGCs that are close to the POA (Supplementary Fig. 8a, b). These results indicate that light-responsive POA neurons are necessary for acute light effect on NREM, but not REM sleep.

To test whether light-responsive POA neurons are also required in daytime sleep, we injected CNO in these same animals at ZT 2. The CNO injection significantly decreased NREM sleep compared to NS injection without influencing REM sleep (Fig. 3g, h). This suggests that light-responsive POA neurons also influence daytime NREM sleep. To determine the role of night-active POA neurons on sleep, we performed the same experiment, but we omitted the 10-hour light pulse (Supplementary Fig. 9a). Similar to the excitatory DREADD results in Supplementary Fig. 7, a few neurons in the POA were labeled (Supplementary Fig. 9b), indicating low numbers that are active at night. CNO injections did not inhibit the nighttime light effects on or daytime levels of NREM or REM sleep (Supplementary Fig. 9c–h). Together, our results show that only inhibiting light-responsive POA neurons blunts the acute effects of light on NREM sleep and lowers daytime NREM sleep.

**Characterizing the light-responsive POA neurons.** We next characterized the cell types of the light-responsive neurons in the POA using genetic labeling combined with single-molecule fluorescence in situ hybridization (smFISH). Fos-2A-iCreER and Ai9 mice received a 10-hour light pulse from ZT 14 to ZT 24 and were injected with 4-OHT at ZT 16 (Fig. 4a). This led to the expression of red fluorescent protein tdTomato only in light-responsive neurons. Four weeks later, smFISH was performed. Corticotropin-releasing hormone (CRH), tachykinin-1 (TAC1), cholecystokinin (CCK), and galanin (GAL) expressing neurons in the POA are implicated in sleep regulation[25,26]. We determined whether the light-responsive neurons are CRH, TAC1, CCK or GAL positive cells. About 90% of the light-responsive neurons are CRH positive, and around 10% express TAC1 (Fig. 4b–g, k). In contrast, none of light-responsive neurons are CCK or GAL positive (Supplementary Fig. 10a–f). More than 90% of the light-responsive POA neurons are labeled with the vesicular GABA transporter (Vgat) (Fig. 4h–j, k), which is a marker of GABAergic neurons. However, none of the POA neurons that are active during baseline dark condition are CRH or TAC1 positive (Supplementary Fig. 11a–g). These findings suggest that light-responsive POA neurons that are GABAergic may be inhibiting wakefulness-promoting neurons in the brain to drive NREM sleep.

**Identification of the downstream targets of light-responsive POA neurons.** To trace the downstream targets of these light-responsive POA neurons, we implemented a selective tracing strategy by using Fos-2A-iCreER mice. In Fos-2A-iCreER mice, Cre-inducible AAV (AAV2/9-phSyn1(S)-FLEX-tdTomato-T2A-SypEGFP-WPRE), which allowed expression of tdTomato under the control of the hSyn promoter and EGFP associated with synaptophysin (Syp), was injected into the POA (Fig. 5a). As we have done previously (Fig. 2), one week later, mice received a light pulse from ZT 14 to ZT 24 and an injection of 4-OHT at ZT 16 to induce the expression of 4-OHT dependent CreER only in light-responsive neurons. Through this strategy, tdTomato will be expressed in the cytoplasm and axons, and EGFP will be localized to the presynaptic terminals of light-responsive POA neurons (Fig. 5b, c). We found that light-responsive neurons in the POA form monosynaptic connections with some of the known downstream targets of the POA, including the tuberomammillary nucleus (TMN), the lateral hypothalamus (LH) (Fig. 5d, g), the ventral tegmental area (VTA) (Fig. 5e, h), and the dorsal raphe nucleus (DRN) (Fig. 5f, i), which are regions well-known for wakefulness promotion. However, these light-responsive neurons in the POA do not innervate other downstream nuclei of the POA, such as the pedunculopontine nucleus (PPT), the laterodorsal tegmental nucleus (LDT) (Supplementary Fig. 12a, d), the parabrachial nucleus (PB) or the locus coeruleus (LC)

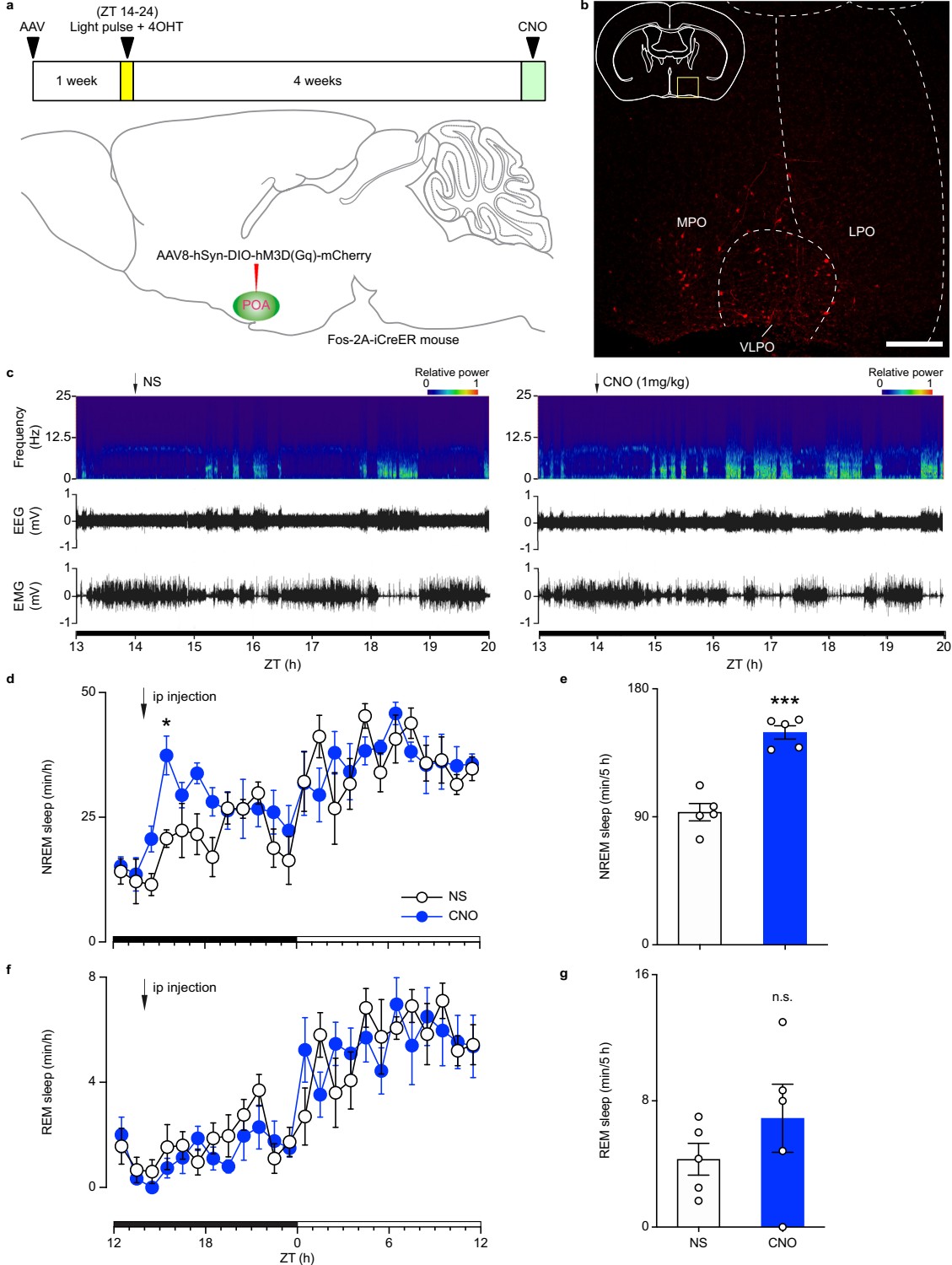

**Fig. 2 Activation of POA light-responsive neurons increased NREM sleep but not REM sleep. a** Schematic showing virus injection, light exposure, 4-Hydroxytamoxifen (4-OHT) injection, and CNO injection. **b** Immunostaining of mCherry in the POA ($n = 5$ animals; scale bar, 200 μm). **c** Typical examples of relative electroencephalogram (EEG) power (normalized by the highest power in each frequency band) and EEG/electromyogram (EMG) traces over 6 h following normal saline (NS) injection (left) or clozapine-N-oxide (CNO) injection (right) at zeitgeber time (ZT) 14. **d** Time course changes in NREM sleep after NS or CNO injection ($n = 5$ animals; two-way ANOVA (ZT 14 to ZT 19), $F_{1, 8} = 53.17$, $P = 0.000085$, Bonferroni post hoc test, *$P < 0.05$). Each cycle represents the hourly mean ± SEM of sleep. The black and white bars on the x axes indicate light-off and light-on periods, respectively. **e** Total time sleep in NREM sleep for 5 h after NS or CNO injection ($n = 5$ animals; two-tailed Student's t test, $t_4 = 11.94$, $P = 0.0003$, ***$P < 0.001$). **f** Time course changes in REM sleep after NS or CNO injection ($n = 5$ animals; two-way ANOVA, $F_{1, 8} = 0.5324$, $P = 0.4864$, Bonferroni post hoc test). **g** Total time sleep in REM sleep for 5 h after NS or CNO injection ($n = 5$ animals; two-tailed Student's t test, $t_4 = 0.8894$, $P = 0.4241$). All error bars denote SEM. MPO, medial preoptic area; LPO, lateral preoptic area; VLPO, ventrolateral preoptic nucleus; n.s., non-significant; NREM, non-rapid eye movement.

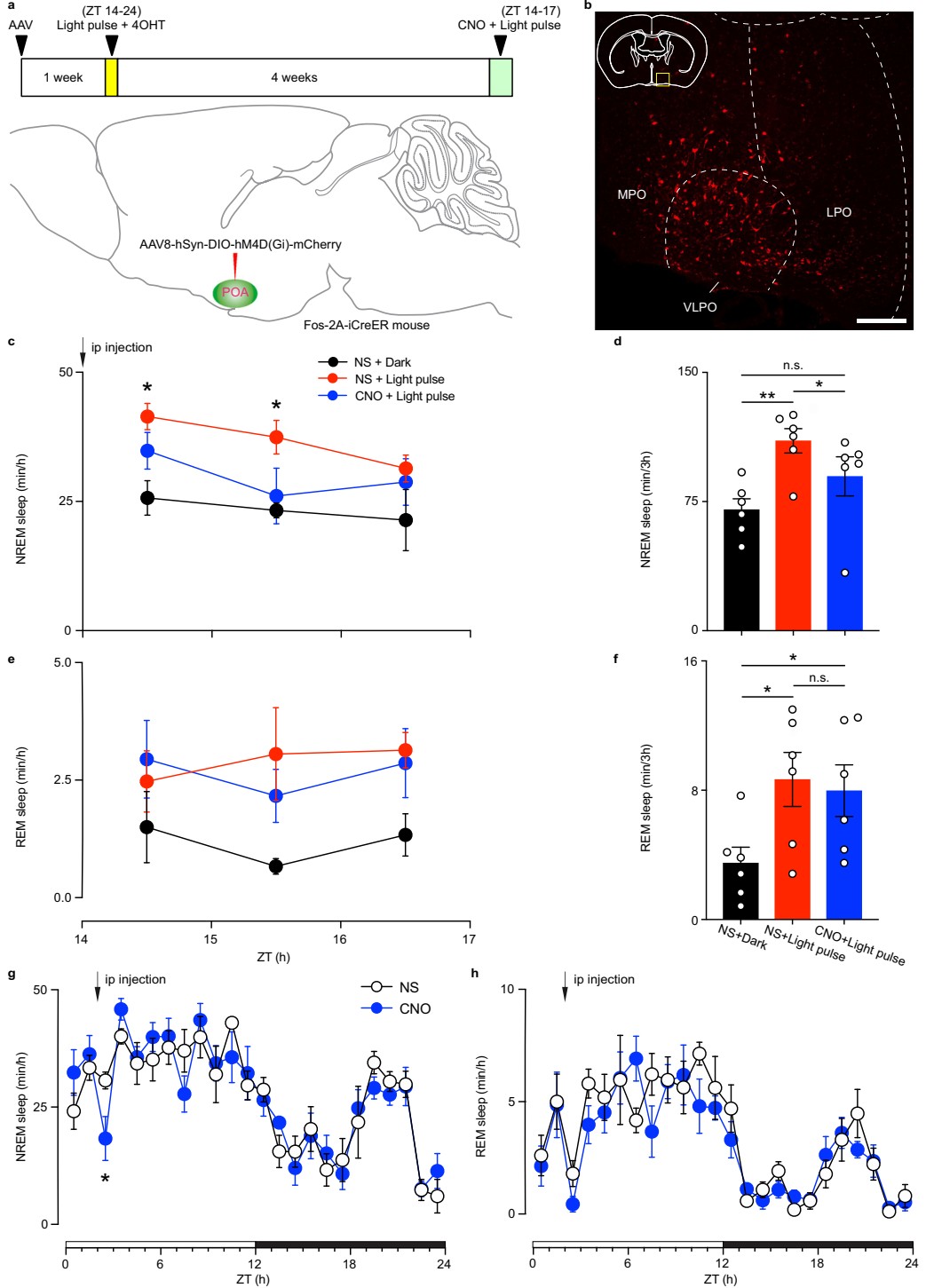

**Fig. 3 Inhibition of POA light-responsive neurons abolished light-induced NREM sleep and spontaneous NREM sleep. a** Schematic showing virus injection, light exposure, 4-Hydroxytamoxifen (4-OHT) injection, and clozapine-N-oxide (CNO) injection. **b** Immunostaining of mCherry in the POA ($n = 6$ animals; scale bar, 200 μm). **c** Time course changes in NREM sleep after normal saline (NS) or CNO injection during light pulse from zeitgeber time (ZT) 14 to ZT 17 ($n = 6$ animals; two-way ANOVA, $F_{2, 15} = 5.453$, $P = 0.0166$, Bonferroni post hoc test. *$P < 0.05$ indicates significant difference between NS + Light pulse and NS + Dark). Each cycle represents the hourly mean ± SEM of sleep. **d** Total time sleep in NREM sleep for 3 h after NS or CNO injection ($n = 6$ animals; one-way ANOVA, $F_{1.351, 6.756} = 18.86$, $P = 0.0026$, Bonferroni post hoc test, *$P < 0.05$, **$P < 0.01$). **e** Time course changes in REM sleep after NS or CNO injection during light pulse from ZT 14 to ZT 17 ($n = 6$ animals; two-way ANOVA, $F_{2, 15} = 3.735$, $P = 0.0483$). Each cycle represents the hourly mean ± SEM of sleep. **f** Total time sleep in REM sleep for 3 h after NS or CNO injection ($n = 6$ animals; one-way ANOVA, $F_{1.194, 5.970} = 7.292$, $P = 0.0328$, Bonferroni post hoc test, *$P < 0.05$). **g** Time course changes in NREM sleep after NS or CNO injection ($n = 6$ animals; two-way ANOVA (ZT 14 to ZT 16), $F_{1, 10} = 6.560$, $P = 0.0283$, Bonferroni post hoc test, *$P < 0.05$). Each cycle represents the hourly mean ± SEM of sleep. The white and black bars on the x axes indicate light-on and light-off periods, respectively. **h** Time course changes in REM sleep after NS or CNO injection ($n = 6$ animals; two-way ANOVA, $F_{1, 10} = 0.9305$, $P = 0.5375$). All error bars denote SEM. MPO, medial preoptic area; LPO, lateral preoptic area; VLPO, ventrolateral preoptic nucleus; n.s., non-significant; NREM, non-rapid eye movement.

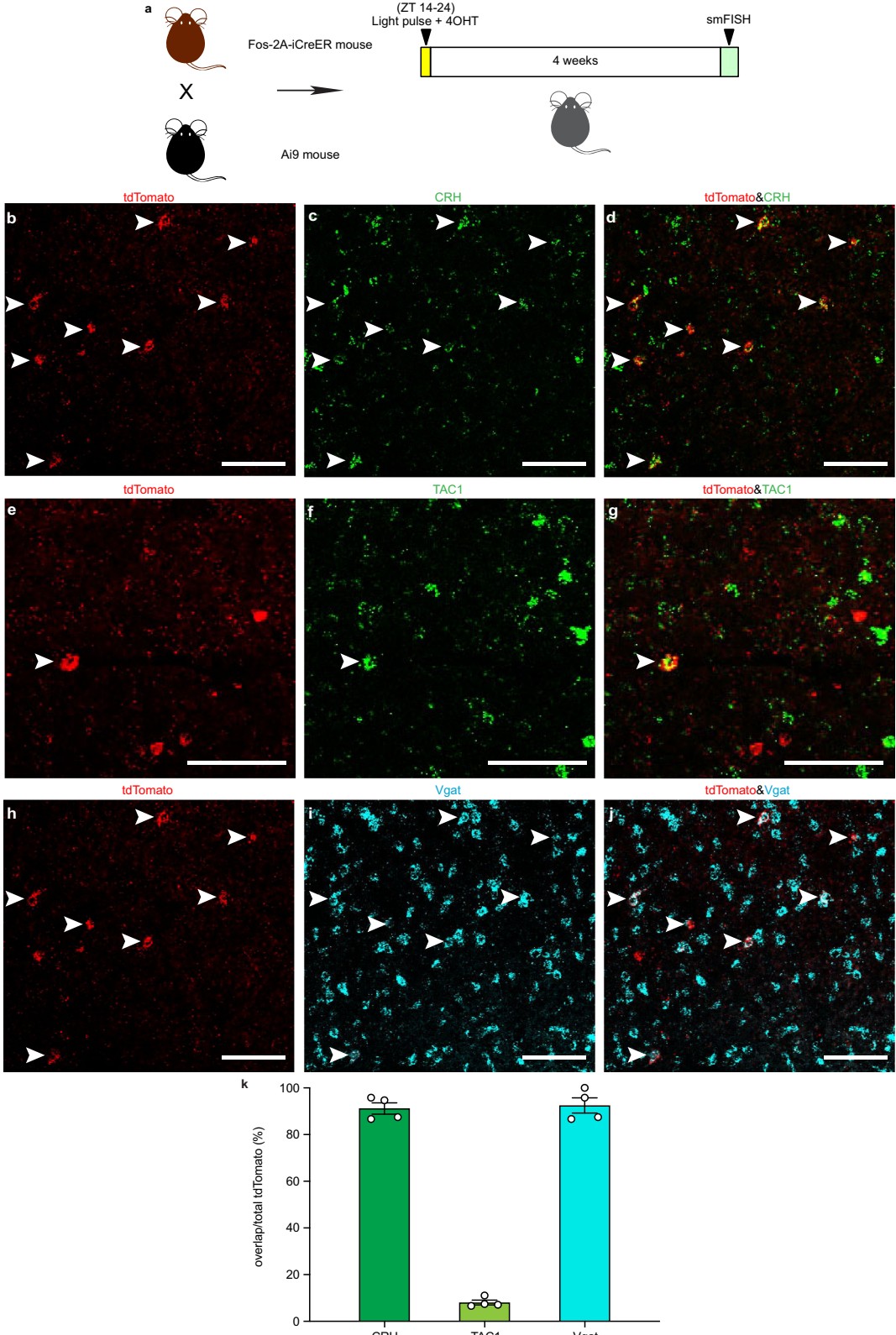

**Fig. 4 Characterization of light-responsive neurons in the POA. a** Schematic showing Fos-2A-iCreER and CAG-FLEX-tdTomato (Ai9) mice that received light exposure, 4-Hydroxytamoxifen (4-OHT) injection and single-molecule fluorescence in situ hybridization (smFISH). **b–d** Distributions of tdTomato positive neurons (**b**), CRH positive neurons (**c**), and colocalization (**d**) in the POA ($n = 4$ animals; scale bar, 100 μm). **e–g** Distributions of tdTomato positive neurons (**e**), TAC1 positive neurons (**f**), and colocalization (**g**) in the POA ($n = 4$ animals; scale bar, 100 μm). **h–j** Distributions of tdTomato positive neurons (**h**), Vgat positive neurons (**i**), and colocalization (**j**) in the POA ($n = 4$ animals; scale bar, 100 μm). Arrowheads indicate colabeled neurons. **k** Percentages of CRH, TAC1, and Vgat positive cells in tdTomato positive neurons ($n = 4$ animals). All error bars denote SEM. ZT, zeitgeber time; CRH, corticotropin-releasing hormone, TAC1, tachykinin-1; Vgat, vesicular GABA transporter.

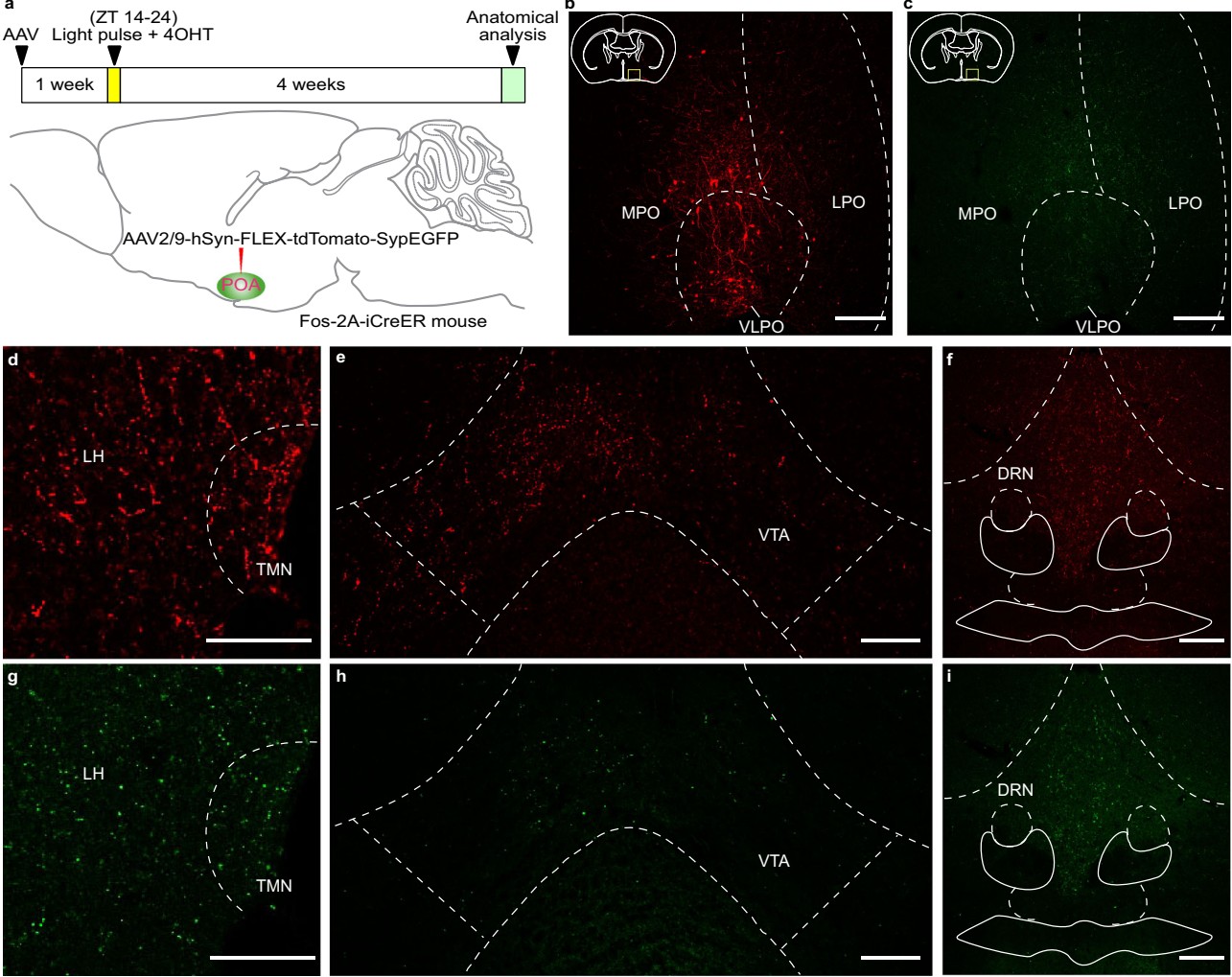

**Fig. 5 POA light-responsive neurons form monosynaptic connections with the TMN, the LH, the VTA, and the DRN. a** Schematic of virus injection, light exposure, and 4-Hydroxytamoxifen (4-OHT) injection. **b, c** Distributions of tdTomato (**b**) and EGFP (**c**) in the POA (*n* = 3 animals; scale bar, 200 μm). **d–f** Distribution of tdTomato in the TMN, the LH (Scale bar, 100 μm) (**d**), the VTA (Scale bar, 100 μm) (**e**), and the DRN (Scale bar, 200 μm) (**f**) (*n* = 3 animals). **g–i** Distribution of EGFP in the TMN, the LH (Scale bar, 100 μm) (**g**), the VTA (Scale bar, 100 μm) (**h**), and the DRN (Scale bar, 200 μm) (**i**) (*n* = 3 animals). ZT, zeitgeber time; POA, preoptic area; MPO, medial preoptic area; LPO, lateral preoptic area; VLPO, ventrolateral preoptic nucleus; TMN, tuberomammillary nucleus; LH, lateral hypothalamus; VTA, ventral tegmental area; DRN, dorsal raphe nucleus.

(Supplementary Fig. 12b, e), some of which are involved in REM sleep regulation[20,34]. In addition, light-responsive POA neurons do not project to the SCN (Supplementary Fig. 12c, f). Thus, we delineated that a subset of ipRGCs project predominantly to CRH-expressing POA neurons to inhibit wakefulness promoting regions in response to acute light stimulations.

## Discussion
In this study, we define a neural circuit, ipRGC-POA, that is both necessary and sufficient for the acute effect of light specifically on NREM sleep (Fig. 6). Our findings suggest that this circuit acts through the inhibition of the TMN, the LH, the VTA and the DRN, areas known for wakefulness-promoting functions[15,25,35–43]. This circuit is separate from the ipRGC-SCN circuit that is involved in circadian photoentrainment of sleep rhythms.

Notably, our results also indicate that there are likely two distinct neural circuits by which light affects NREM versus REM sleep, as acute light exposure in wild-type animals affects NREM and REM sleep[12,16]. This implies that light can play an intricate role in regulating sleep through three parallel pathways: The ipRGC-SCN for photoentrainment, the ipRGC-POA for acute

effect on NREM sleep, and a yet to be discovered pathway from ipRGCs that regulates REM sleep.

Interestingly, the duration of the effects of CNO by the excitatory DREADD on NREM sleep were different when ipRGCs in the retina were activated compared to those of the light-activated POA neurons in the brain. When ipRGCs that project to the POA are activated by CNO, the effects on NREM sleep last for 2 h (Fig. 1j), whereas when the light-responsive POA neurons are activated by CNO, the NREM sleep effects last for 5 h (Fig. 2d). These differences may arise due to adaptation in the available vesicles for neurotransmitter release from ipRGC terminals or alternatively, adaptation in the postsynaptic receptors of the POA neurons that receive ipRGC input could underlie these differences.

Moreover, we demonstrated that the light-responsive POA neurons constitute predominantly one of the four different cell subpopulations in the POA that influence sleep[25]. These light-responsive POA neurons are mostly CRH expressing cells, which are GABAergic and form inhibitory monosynaptic connections with wakefulness-promoting brain regions.

In addition, our data explain a curious finding we observed in 2008, where we showed that although light affects motor activity

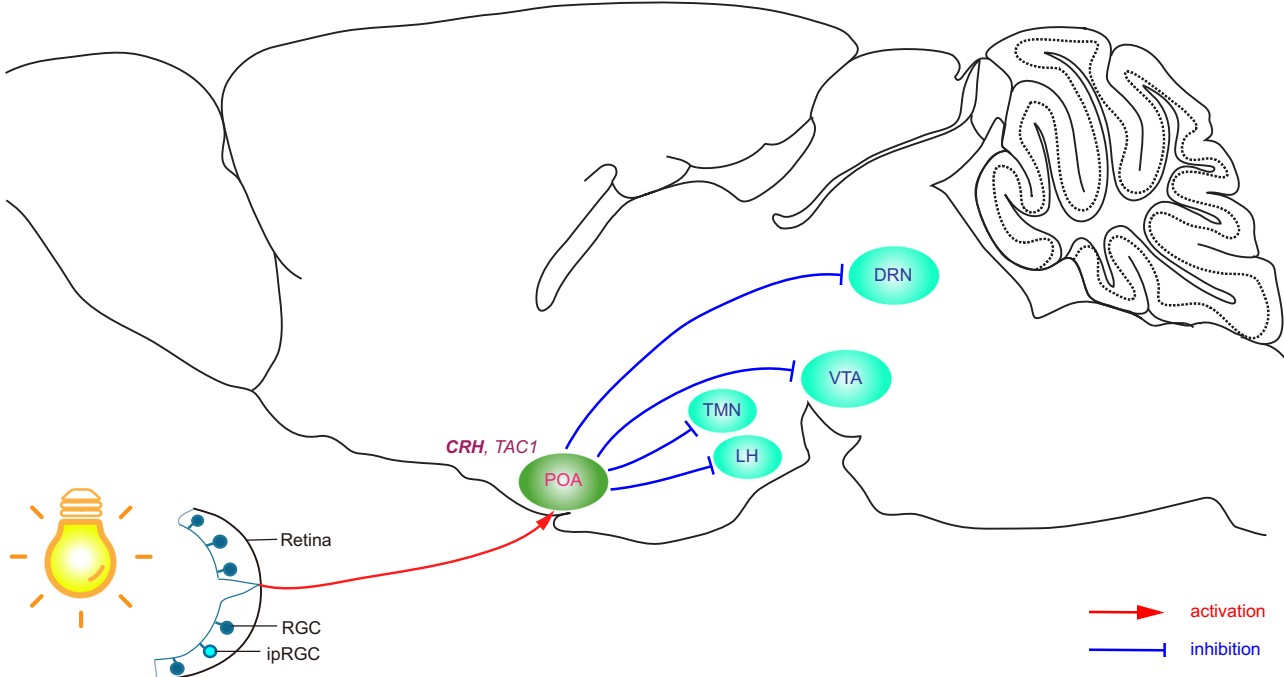

**Fig. 6 Schematic neural circuit of acute light effect on sleep in mice.** Acute light stimulates ipRGCs, which directly activate majorly CRH and to a lesser extent TAC1 expressing GABAergic neurons in the POA. These GABAergic POA neurons form inhibitory monosynaptic connections with wakefulness-promoting regions, including the TMN, the LH, the VTA, and the DRN. POA, preoptic area; TMN, tuberomammillary nucleus; LH, lateral hypothalamus; VTA, ventral tegmental area; DRN, dorsal raphe nucleus.

for every dark to light transition, the story is different for sleep[11]. Specifically, there are dark to light transitions where light fails to acutely induce sleep. The fact that ipRGCs only regulate one of the three populations in the POA allows the other two populations to drive sleep when light input becomes aberrant or the homeostatic drive becomes dominant.

Surprisingly, we found that the light-responsive neurons in the POA only innervate some of the downstream nuclei of the POA, including the TMN, the LH, the VTA and the DRN. The density of the fibers of light-responsive POA neurons in these downstream targets is similar to the published report[36]. However, we find no projections from light-responsive POA neurons to the PPT, the LDT, the PB or the LC, in contrast to the non-specific tracing of the POA[36]. Significantly, the LC is thought to suppress REM sleep[20,34]. Therefore, the lack of innervations from POA light-responsive neurons to the LC may account for the incapacity of ipRGC-POA circuit influencing REM sleep.

Nighttime light exposure induces wakefulness in diurnal animals including humans in contrast to sleep in nocturnal species. This highlights neural circuit or molecular differences between diurnal and nocturnal animals downstream of the light input. These differences may be located in the retina and/or the brain. It is intriguing to speculate whether ipRGCs that express GABA[44] may have inhibitory role on POA neurons in diurnal species. For brain mechanisms, the projection patterns of light-responsive POA in diurnal animals may differ from those of nocturnal animals. More mechanistic studies are needed to reveal the locus of divergence between diurnal and nocturnal animals in their responses to acute light.

Together, our finding indicates that an ipRGC-POA circuit is both necessary and sufficient for the acute effect of light on NREM sleep, independent of the circadian pacemaker.

## Methods

**Mice**. All animal care and experimental procedures were approved by the Animal Care and Use Committees of the National Institute of Mental Health. All efforts were made to minimize the pain and the number of animals used. Fos-2A-iCreER (Stock # 030323) and CAG-FLEX-tdTomato (Ai9) (Stock # 007909) mice were obtained from the Jackson Laboratory. $Opn4^{Cre}$ mice used in this study were obtained by mating the $Opn4^{Cre}$ mice that were generated in our lab using homologous recombination in embryonic stem cells and replacing the melanopsin gene with the Cre recombinase[30]. Wild-type mice (Stock # 101043) were obtained from the Jackson Laboratory. 6-8 weeks old male mice were used at the start of the experimental procedures. Mice had access to food and water ad libitum and were maintained at constant ambient temperature (21–23 °C), humidity (40–60%), and 12 h light/dark cycle (100 lux, light on at 06:00 am). Animals with telemetries were housed individually.

**Intravitreal injection**. Mice were anesthetized using isoflurane and placed under a stereomicroscope. The microscope and all the instruments were properly cleaned and sterilized. A glass needle (pulled 10 µl microcapillary tube, Sigma P0674) was used to drive AAV2-CAG-DIO-GFP (2 µl, Salk Institute Viral Vector Core) or AAV2-Ef1α-DIO-Flp (2 µl, Salk Institute Viral Vector Core) into the vitreous chamber of the eye to ensure delivery specifically to the retina. After slowly injecting the total volume, pipette was left in place for 60–90 s. Mice recovered from injections on a heating pad until they woke from anesthesia.

**Surgery and virus injection**. The stereotaxic frame and all the instruments were properly cleaned and sterilized. Mice were deeply anesthetized using isoflurane; fully unconscious was confirmed by complete absence of flinching response to pinch. Skull fur was properly shaved, the head of the mouse was then fixed to the stereotaxic frame, cleaned by scrubbing with povidone-iodine and 70% ethanol. The skull was exposed using a sterile scalpel. A small hole was drilled over the region of interest. All coordinates used refer to the mouse brain atlas[45]. 100 nl AAVretro-Ef1α-fDIO-hM3D(Gq)-EGFP (Biohippo Inc.) or AAVretro-hSyn-DIO-EGFP (Addgene) mixed with AAV8-hSyn-mCherry (Addgene), AAV8-hSyn-DIO-hM3D(Gq)-mCherry (Addgene), AAV8-hSyn-DIO-hM4D(Gi)-mCherry (Addgene), or AAV2/9-phSyn1(S)-FLEX-tdTomato-T2A-SypEGFP-WPRE (Boston Children's Hospital Viral Core) was slowly injected (40 nl/min) into the POA (AP = +1.0 mm; ML = ±0.6 mm; DV = −5.1 mm). The glass pipette was left in place for eight additional minutes and then slowly withdrawn.

Three weeks after injections, mice that underwent polysomnographic recordings were implanted with electrodes for electroencephalogram (EEG) and electromyogram (EMG) recordings under isoflurane anesthesia. The abdominal implanted telemetry (Model HD-X02, DSI) consists of two EEG collecting wires connected with two stainless steel screws (1.19 mm in diameter) as EEG electrodes, which were inserted through the skull (+1.0 mm anteroposterior; +1.5 mm mediolateral from bregma or lambda) according to the mouse brain atlas[45], and two EMG electrodes that were placed into neck musculature. The EEG electrodes

were fixed to the skull with dental acrylic integrity temporary crown and bridge material (Henry Schein Animal Health). Mice were individually housed in recording systems after surgery.

**Intraperitoneal injection**. All the intraperitoneal injections during the dark period were performed under red light at 8 lx, which does not influence sleep in mice[16]. CNO dose was 1 mg/kg body weight for activation in Gq-expressing mice and inhibition in Gi-expressing mice.

**Polysomnographic recording and analysis**. After 10-day recovery and acclimation period, EEG and EMG were recorded with Ponemah software (DSI). The sleep-wake states were automatically classified 10 s periods as wakefulness, NREM sleep, or REM sleep with the NeuroScore software (DSI). The automatically defined wake or sleep stages were then checked visually and corrected if necessary.

**Immunohistochemistry**. Mice were deeply anaesthetized with isoflurane and transcardially perfused with 0.1 M PBS followed by 4% paraformaldehyde (w/v) in PBS. For fixation, eyes and brains were kept in 4% paraformaldehyde (1.5 h and overnight, respectively). Retinas were isolated and washed three times for 15 min each in 0.1 M PBS. Then retinas were incubated in 0.1 M PBS with 5% donkey serum (Vector Labs) and 0.3% Triton X-100 (Sigma–Aldrich) at 4°C overnight. Retinas washed three times for 15 min each in 0.1 M PBS and incubated with primary antibody in 0.1 M PBS containing 0.3% Triton X-100 and 5% donkey serum at 4°C for 3 days (anti-GFP, ab13970, 1:2000, Abcam; anti-RFP, LS-C340696, 1:1000, LSBio; anti-OPN4, AB-N38, 1:1000, Advanced Target Systems). Primary antibodies were washed three times in 0.1 M PBS before incubation with secondary antibody overnight at 4°C (Alexa 488-AffiniPure donkey anti-chicken, 703-545-155, 1:500, Jackson ImmunoResearch; Alexa 594-conjugated donkey anti-goat, A-11058, 1:500, Thermo Fisher Scientific; Alexa 647-conjugated donkey anti-rabbit, A-31573, 1:500, Thermo Fisher Scientific). Brains were incubated in 0.1 M PBS containing 20% sucrose until they sunk to the bottom. Coronal sections (40 μm) were cut using a cryostat (Leica). Brain sections were incubated in 0.1 M PBS with 5% donkey serum and 0.3% Triton X-100 for 2 h. Then the brain sections were washed five times for 3 min each in 0.1 M PBS and incubated with primary antibody in 0.1 M PBS containing 0.3% Triton X-100 and 5% donkey serum at 4°C overnight (anti-GFP, ab13970, 1:2000, Abcam; anti-RFP, LS-C340696, 1:1000, LSBio). Primary antibodies were washed five times for 3 min each in 0.1 M PBS before incubation with secondary antibody for 2 h (Alexa 488-AffiniPure donkey anti-chicken, 703-545-155, 1:500, Jackson ImmunoResearch; Alexa 594-conjugated donkey anti-goat, A-11058, 1:500, Thermo Fisher Scientific).

**smFISH**. Samples were collected using a flash frozen method. Isopentane (Sigma–Aldrich) was chilled on dry ice for 30 m. The mice were deeply anaesthetized with isoflurane and the brain was dissected quickly then placed in the chilled isopentane for 20 s. Frozen samples were kept at −80 °C overnight. Sections were cut 16 μm thick using a cryostat and mounted. Brain sections were then fixed using chilled 4% paraformaldehyde for 20 m and rinsed twice in 0.1 M PBS for 1 m each. The sections were incubated in 50%, 70%, and 100% ethanol for 5 m each, then stored in 100% ethanol at −20 °C overnight. The next day, smFISH was performed using RNAscope fluorescent multiplex assays (Advanced Cell Diagnostics). Sections were incubated at room temperature in Protease IV for 20 m then rinsed twice in deionized H$_2$O for 1 m each. Sections were incubated with probes for 2 h at 40 °C in a HybEZ oven. The probes were decanted, and the sections were rinsed twice in 1x wash buffer (50x wash buffer diluted in deionized H$_2$O) for 2 m each. Samples were incubated in the 40 °C oven with AMP 1 (30 m), AMP 2 (15 m), AMP 3 (30 m), and AMP 4 Alt B (15 m). Between each amplification step, the sections were rinsed in 1x wash buffer twice for 2 m each. The slides were counterstained with DAPI and cover slipped. Probes used for RNAscope (Advanced Cell Diagnostics): tdTomato (C1), CRH (C2), TAC1 (C2), CCK (C2), GAL (C2), Vgat (C3). Images of sections containing the POA (bregma +1.24 mm ~ −0.64 mm) were obtained. Numbers of different cell populations were counted using ImageJ. Results obtained from three brain sections were averaged in each mouse.

**Image analysis**. Fluorescent images were collected with an Eclipse Ti2 confocal microscope (Nikon). Images were overlaid and visualized using ImageJ software.

**Statistical analysis**. All data are expressed as mean ± SEM. Comparisons of time course changes in sleep of control and experimental mice were performed using two-way ANOVA with Geisser-Greenhouse correction followed by a Bonferroni post hoc test. Comparisons of sleep amounts or power density between control and experimental groups were evaluated using a paired, two-tailed Student's t-test. Prism 8.0.2 (GraphPad Software) was used for all statistical analyses. In all cases, $p < 0.05$ was considered to be significant.

**Reporting summary**. Further information on research design is available in the Nature Research Reporting Summary linked to this article.

## Data availability
The sleep and histology data generated in this study are provided in the Supplementary Information/Source Data file. Source data are provided with this paper.

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

## Acknowledgements
We thank the members of Section on Light and Circadian Rhythms at National Institute of Mental Health, and Johns Hopkins Biology Mouse Tri-Lab for helpful discussions. We are particularly grateful for Dr. Haiqing Zhao (Johns Hopkins University) for the rigorous reading of the manuscript. We also thank the NIMH IRP Rodent Behavioral Core for providing the stereotaxic system. This work was supported by the intramural research at the National Institute of Mental Health (ZIAMH002964-02).

## Author contributions
Z.Z. and S.H. designed the experiments and wrote the manuscript; Z.Z., C.B. and T.W. performed experiments. Z.Z. performed data analysis.

## Competing interests
The authors declare no competing interests.
