## [Peer Review File · Nature Communications]

The retinal ipRGC-preoptic circuit mediates the acute effect of light on sleepREVIEWER COMMENTS

Reviewer #1 (Remarks to the Author):

In this paper, the authors use a series of innovative chemogenetic studies to investigate the role of melanopsin ipRGC projections in the regulation of sleep. Despite the interest in the regulation of sleep by light, the specific neural pathways mediating this response and the role of melanopsin are poorly understood. As such, this paper is an extremely important contribution to this field, convincingly demonstrating that a specific ipRGC projection to a CRH positive subset of POA neurons plays a key role in sleep regulation by inhibiting wake promoting nuclei. This is a major advance in this field and will be of great interest to the broader scientific community.

Overall, the paper is very clearly written and the experiments all progress logically. The data are convincing and are all clearly presented, using an elegant and well-thought out combination of chemogenetic and transgenic tools. My comments below primarily relate to a few simple areas that could be addressed to help clarify the findings.

MAIN COMMENTS

1. The effects of CNO on NREM sleep are clear-cut but relatively small. For example, in Fig 1j, this is an increase of ~20 minutes of NREM sleep in 2h (or ~10 mins/h). In Fig 2e, this is ~60mins in 5 hours (~12 mins/h). Why is a different time period used here? Some justification would help. Furthermore, in previous papers on light-induced sleep, the amount of sleep per 30 min bin increases from ~35% to 80% (~27 mins/h) in Altimus et al., 2008. In another study, sleep per 1h bin increases from ~10% to ~60% (~30 mins/h). It would be good to discuss the differences in chemogenetically- vs light-induced sleep. Could these be due to incomplete ipRGC transduction? Due to rod/cone inputs to ipRGCs? Or additional pathways that normally contribute to sleep induction? (for example, the effects of light on activity may facilitate sleep).

2. In several experiments the authors use a 10h light pulse from ZT14-24 to identify light responsive POA neurons. This is an interesting and elegant approach. Was activity or sleep behaviour confirmed during this time course? If mice have been under a normal 12:12 LD cycle, they will have been predominantly asleep prior to this light pulse. As such, homeostatic sleep pressure will be low, and animals may not be asleep throughout this period of light exposure. This may potentially result in both sleep and wake active neurons in the POA being active and transduced.

3. Quantification of colocalization. In Fig 3, images of colocalization of CRH, TAC1 and CCK with tdTomato is shown. The authors conclude that CRH neurons are predominantly CRH positive, but no quantification is provided to support this statement. There are clearly limitations of quantification in this manner (and this may be affected via the use of smFISH), but it would be good to know the extent of this neuronal specificity. E.g. is this >90% of tdTomato marked neurons that are CRH positive? How much less is the colocalization with TAC1 and CCK?

MINOR POINTS

1. The conclusions on the effects on REM sleep could benefit from a little more justification. Given the relatively small change in total sleep observed, the amount of REM sleep that is likely to occur will be extremely small. As such, there may be insufficient REM sleep occurring to really conclude that that this is entirely unaffected. This may affect the conclusion regarding NREM- and REM-specific pathways.

2. POA neurons are known to be GABAergic, inhibiting sleep promoting centres. This is the basis of the flip-flop model proposed by Saper (e.g. Saper, Scammell and Lu, 2005 Nature). As such, whilst this conclusion is clearly scientifically solid and mechanistically interesting, it is perhaps not remarkable as is suggested.

3. How do the observed projections of POA neurons in this study correspond to the density of known projections? Could the observed pattern of projections simply reflect the strongest projections of the POA, rather than a specific pattern of projections to wake-promoting centres?

Reviewer #3 (Remarks to the Author):

This work identified a novel circuit (ipRGC-POA) by which acute light exposure increases sleep. Importantly, this circuit is independent of the well-known connection between ipRGC and the SCN. Using genetic tracing and chemogenetically-assisted circuit dissection, the authors show that a novel subset of light-responsive Vgat⁺ and CRH⁺ neurons in the POA increase NREM sleep (but not REM sleep) by monosynaptic inhibition of arousal promoting systems that include the TMN, lateral hypothalamus, VTA and DRN.

The authors conclude that this pathway may "provide a possible target for drug development to treat sleep disorders caused by night-time light exposure."

The study is highly relevant and timely because it advances our understanding of the complex role of the preoptic area in the regulation of sleep and wakefulness. The experiments are well designed, and the data are neat and clearly presented. Interpretations and conclusions are appropriate.

I have the following comments/questions for the authors' consideration:

- 1) In general, it is fair to say that light causes arousal in diurnal species. The role and relevance of the light-induced increase in NREM sleep in mice requires some discussion. Do these cells play any role in sleep generation during the daytime?
- 2) Tracing pathways from ipRGC clearly spare the SCN. Did the authors verify -or is there any evidence that suggests- if light-sensitive POA neurons innervate the SCN?
- 3) Looks like these cells are contained within a large area of the preoptic region. Are these POA neurons localized within specific anatomical areas/nuclei. Can you be more specific, including an approximate AP range.
- 4) The authors report dense POA innervation by ipRGC. This is not accurately reflected by Figure 1c. Perhaps they can provide a better image that unequivocally represent the group data.
- 5) The effect of the activation of light-sensitive POA neurons on NREM sleep is clear, but modest. It would be interesting to compare the quality of NREM sleep by assessing EEG power during NREM sleep between conditions.
- 6) The effect stimulation of night active neurons on REM sleep is very mild (Suppl Figure 3) and too distant from the time of injection. It does not look like a clear drug/stimulation effect, and does not support the interpretation that these cells influence REM sleep. Additionally, very few neurons were labeled, which raises the question whether it was the scarce number of cells that was not enough to increase NREM sleep. The number of neurons that expressed hM3 receptors in comparison with the other conditions.
- 7) Relative to point #2, Lu et al., 2000 (PMID: 10804223) described a subset of preoptic neurons that expressed Fos during REM sleep. Based on location, they refer to this cluster as "extended VLPO". Can you tell if light-sensitive POA neurons are present within this area?
- 8) What was the bin length used for sleep scoring? Please provide this info.
- 9) It took me a while to see that the SCN and SON in Suppl Figure 2 do not have mCherry expression. Delineating the boundaries of both structures as in Suppl Fig. 4 would make it more evident.
- 10) Was mCherry not enhanced by immunohistochemistry?
- 11) Please provide the number of mice used in experiments summarized by Figures 3 and 4.

Reviewer #4 (Remarks to the Author):

In this article, Zheng and colleagues report that melanopsin (OPN4) neuronal projections to the preoptic area mediate the sleep-inducing effects of light in nocturnal rodents and that the POA neurons activated by light are CRH⁺ or Tac1⁺. These findings are novel, and they assume great significance as the light is the primary cue that determines the timing and amounts of sleep. Overall, the manuscript is well-written, and the figures are presented clearly. However, I have significant concerns including the sufficiency of datasets for the conclusions derived. In addition, inadequate authentication of methods and the lack of adequate controls largely reduce the enthusiasm. Nevertheless, I believe that the manuscript, when appropriately revised, will be of

great interest to the sleep and circadian biology community.

1. Crucial controls are not in place (WT mice controls/AAV controls).
2. Selectivity and specificity of the AAVs as well as their transduction efficiency was not reported.
3. From the figure 1, ipRGC(OPN4)-projections to the POA appear to be restricted to the ventrolateral preoptic region (VLPO); but, do these neurons also project to other POA regions? Figure (s) showing the entire POA and at least 2-3 rostrocaudal levels should be included, which will be helpful to visualize the projection pattern more clearly. If the projections are restricted to the POA, why not use the term 'VLPO'? Apart from POA, where else do these neurons project? How do the authors rule out the involvement of other projections in light-induced sleep?
4. Supplementary fig 1 a, b – Although no viral expression is observed in the SCN or SON, significant expression is seen in the subparaventricular zone (SPZ), which also receives retinal projections. Did the authors rule out the contribution of SPZ for the observed effects? Only the cases without significant involvement of SPZ should be included or anatomical controls with injections in the SPZ should be presented.
5. Are there any specific reason why the authors chose ZT14 (rather than ZT12, for e.g.) for injections? How were these injections during the dark period performed? If red light was used, the intensity of the light levels may be reported? Even red light above a certain threshold has been shown to modify animal behavior.
6. Activation of retinal projections to POA and light pulse-trapped POA neurons promoted SWS. But it is very important to show if inhibition or loss of these neurons prevents the light-induced changes in sleep, without which it is difficult assign a causal role for these neurons in this function. Similarly, several subsets of neurons in the POA may be sleep-active, which also may be directly or indirectly activated by light. Are the OPN neurons specifically target the CRH/Tac1 cells in the POA? Are these neurons selectively mediate the photic effects on sleep or they are part of sleep-circuitry on which light acts to promote sleep?
7. As the light exposure increases sleep, light pulse-trapped neurons (cFos-trapped) could just be sleep-active neurons in the POA. As described in the manuscript, sleep in the dark-period is low; thus, activation of fewer POA sleep-active neurons might not be able to induce more sleep than occurs during the dark-period. This raises a concern if the 'no-light pulse' controls are sufficient to distinguish 'sleep-active' neurons from 'light-activated' neurons? This also raises another concern if the 'night-activated' neurons were negative for CRH or Tac1? smFish data of night-activated neurons are not presented.
8. What percentage of light-responsive neurons in the POA are CRH or Tac1+? As this is not reported, it leads to another question if any of these neurons are galanin+? In addition to CRH, Tac1 and CCK, galaninergic neurons in this region has been associated with sleep for a long time.
9. The telemetry transmitter also provides body temperature data. Did the authors observe any changes in body temperature after activation of either ipGRCs or POA cells? Several populations of POA neurons may alter both sleep and temperature and these variables strongly influence each other.

Point by point rebuttal to reviewers' comments

For clarity, our responses to the reviewers' comments are in blue.

We thank the reviewers for the excellent review of our original manuscript. Thanks to their suggestions and the new experiments we carried out, we believe our conclusions are much more strengthened in the revised manuscript.

Reviewer #1 (Remarks to the Author):

In this paper, the authors use a series of innovative chemogenetic studies to investigate the role of melanopsin ipRGC projections in the regulation of sleep. Despite the interest in the regulation of sleep by light, the specific neural pathways mediating this response and the role of melanopsin are poorly understood. As such, this paper is an extremely important contribution to this field, convincingly demonstrating that a specific ipRGC projection to a CRH positive subset of POA neurons plays a key role in sleep regulation by inhibiting wake promoting nuclei. This is a major advance in this field and will be of great interest to the broader scientific community.

Overall, the paper is very clearly written and the experiments all progress logically. The data are convincing and are all clearly presented, using an elegant and well-thought out combination of chemogenetic and transgenic tools. My comments below primarily relate to a few simple areas that could be addressed to help clarify the findings.

We truly appreciate the reviewer's positive comments on the novelty and significance of our study.

MAIN COMMENTS

1. The effects of CNO on NREM sleep are clear-cut but relatively small. For example, in Fig 1j, this is an increase of ~20 minutes of NREM sleep in 2h (or ~10 mins/h). In Fig 2e, this is ~60 mins in 5 hours (~12 mins/h). Why is a different time period used here? Some justification would help.

We apologize for the confusion in presenting the data. In the original manuscript Fig. 1i (revised manuscript Fig. 1j), when ipRGCs that project to the POA are activated by CNO, the NREM sleep effects last for 2 hours, whereas, in original manuscript Fig. 2d (revised manuscript Fig. 2d), when the light-responsive POA neurons are stimulated by CNO, the NREM sleep effects last for 5 hours. This is simply why we chose these durations to analyze the effects on NREM sleep. Now we make this clearer in the revised manuscript.

We also add a new section in the discussion of the revised manuscript to speculate on why activating ipRGCs may lead to shorter effects on sleep than activating POA neurons directly. Here is the section: “Interestingly, the duration of the effects of CNO by the excitatory DREADD on NREM sleep were different when ipRGCs in the retina were activated compared to those of the light-activated POA neurons in the brain. When ipRGCs that project to the POA are activated by CNO, the effects on NREM sleep last for 2 hours (Fig. 1j), whereas, when the light-responsive POA neurons are activated by CNO, the NREM sleep effects last for 5 hours (Fig. 2d). These differences may arise due to adaptation in the available vesicles for neurotransmitter release from ipRGC terminals or alternatively, adaptation in the postsynaptic receptors of the POA neurons that receive ipRGC input could underlie these differences.”

Furthermore, in previous papers on light-induced sleep, the amount of sleep per 30 min bin increases from ~35% to 80% (~27 mins/h) in Altimus et al., 2008. In another study, sleep per 1h bin increases from ~10% to ~60% (~30 mins/h). It would be good to discuss the differences in chemogenetically- vs light-induced sleep. Could these be due to incomplete ipRGC transduction? Due to rod/cone inputs to ipRGCs? Or additional pathways that normally contribute to sleep induction? (for example, the effects of light on activity may facilitate sleep).

We thank the reviewer for this important comment. The increase in total sleep in *Altimus et al.*, 2008 included both NREM and REM sleep (shown in Fig. 1B). The total sleep increase for the 3-hour period is from ~35% to 65%, which is ~18 mins/h. In our study, only NREM sleep was affected and the effects were ~12 mins/h, which is not much different from the *Altimus et al.*, 2008. The reviewer used the numbers in the first 1.5 h in Fig. 1C of *Altimus et al.*, 2008 to

indicate a change of ~35% to 80%, which comes out to be ~27 mins/h and looks quite different from our current data.

More importantly, we now carry out the same type of experiments to those of *Altimus et al.*, 2008 in the current study. We recorded NREM and REM sleep changes separately as shown in our revised manuscript Fig. 3c-f. We found that 3h acute light pulse increased NREM sleep from 70 mins to 110 mins, which is ~13 mins/h. These data indicate that the increase of NREM sleep by chemogenetic activation is similar in magnitude to those induced by light.

2. In several experiments the authors use a 10h light pulse from ZT14-24 to identify light responsive POA neurons. This is an interesting and elegant approach. Was activity or sleep behaviour confirmed during this time course? If mice have been under a normal 12:12 LD cycle, they will have been predominantly asleep prior to this light pulse. As such, homeostatic sleep pressure will be low, and animals may not be asleep throughout this period of light exposure. This may potentially result in both sleep and wake active neurons in the POA being active and transduced.

This is again an insightful point by this reviewer. We have not recorded the activity or sleep during the 10-h light pulse, however, importantly, our data shows that different subpopulations in the POA are labeled during the 10-h light pulse versus the baseline dark condition. We now provide this data in our revised manuscript as Supplementary Fig. 10. This new supplementary figure should be compared to Fig. 4 of the revised manuscript.

Briefly, our smFISH data show that the subpopulations in the POA that we labeled during 10-h light pulse are CRH (~90%) or TAC1 (~10%) positive, whereas those we labeled during baseline dark condition are not labeled at all with either CRH or TAC1. These results provide strong evidence that the 10-h light pulse activated neurons are distinct from dark controls.

3. Quantification of colocalization. In Fig 3, images of colocalization of CRH, TAC1 and CCK with tdTomato is shown. The authors conclude that CRH neurons are predominantly CRH positive, but no quantification is provided to support this statement. There are clearly limitations

of quantification in this manner (and this may be affected via the use of smFISH), but it would be good to know the extent of this neuronal specificity. E.g. is this >90% of tdTomato marked neurons that are CRH positive? How much less is the colocalization with TAC1 and CCK?

As suggested, we now quantify this data as shown in revised manuscript Fig. 4k. We find that 90% of tdTomato neurons are CRH positive and 10% of tdTomato neurons are TAC1 positive. Remarkably, we generated new data showing that the tdTomato neurons do not colocalize with CCK or GAL (revised manuscript Supplementary Fig. 9).

MINOR POINTS

1. The conclusions on the effects on REM sleep could benefit from a little more justification. Given the relatively small change in total sleep observed, the amount of REM sleep that is likely to occur will be extremely small. As such, there may be insufficient REM sleep occurring to really conclude that that this is entirely unaffected. This may affect the conclusion regarding NREM- and REM-specific pathways.

Although the reviewer raises a good point, we do not think that this is the cause for the lack of effects on REM sleep. In fact, to further study the role of ipRGC-POA circuit on the acute light effects on NREM and REM sleep, we now label the light-responsive POA neurons with inhibitory DREADDs hm4D(Gi) and study the effects of a light pulse given from ZT 14 to ZT 17 that is known to affect both NREM and REM sleep. We injected NS or CNO at several minutes before ZT 14 and then we applied the light pulse. In NS injected group, both NREM and REM sleep were significantly increased compared to dark control as shown in revised manuscript Fig. 3c-f. In CNO injected group, however, only the effects of light on NREM sleep were blunted as shown in revised manuscript Fig. 3c-f. The lack of inhibition on REM sleep by the CNO (inhibitory DREADDs) suggests specificity of the ipRGC-POA in its influence on NREM sleep.

2. POA neurons are known to be GABAergic, inhibiting sleep promoting centres. This is the basis of the flip-flop model proposed by Saper (e.g. Saper, Scammell and Lu, 2005 Nature). As

such, whilst this conclusion is clearly scientifically solid and mechanistically interesting, it is perhaps not remarkable as is suggested.

In *Kroeger et al., 2018* study, Saper's group reported that GAL neurons in the VLPO play a critical role in sleep regulation. However, in our study, we found that the light-responsive POA neurons belong to CRH/TAC1 subpopulations, but not the GAL neurons (revised manuscript Supplementary Fig. 9). This indicates that the cell populations in the POA that Saper proposed for the flip-flop model are distinct from what we found in our study.

3. How do the observed projections of POA neurons in this study correspond to the density of known projections? Could the observed pattern of projections simply reflect the strongest projections of the POA, rather than a specific pattern of projections to wake-promoting centres?

The density of the fibers of light-responsive POA neurons in the TMN, the LH, the VTA, and the DRN is similar to the published report (Sherin et al., 1998). However, we find no projections from light-responsive POA neurons to the PPT, the LDT, the PB or the LC (included as a new figure in our revised manuscript Supplementary Fig. 11), in contrast to the non-specific tracing of POA neurons (Sherin et al., 1998). Because of the great suggestion by this reviewer, we now discuss the implications of this differential input to distinct brain regions: "Surprisingly, we found that the light-responsive neurons in the POA only innervate some of the downstream nuclei of the POA, including the TMN, the LH, the VTA and the DRN. The density of the fibers of light-responsive POA neurons in these downstream targets is similar to the published report (Sherin et al., 1998). However, we find no projections from light-responsive POA neurons to the PPT, the LDT, the PB or the LC, in contrast to the non-specific tracing of the POA (Sherin et al., 1998)."

Reviewer #3 (Remarks to the Author):

This work identified a novel circuit (ipRGC-POA) by which acute light exposure increases sleep. Importantly, this circuit is independent of the well-known connection between ipRGC and the SCN. Using genetic tracing and chemogenetically-assisted circuit dissection, the authors show

that a novel subset of light-responsive *Vgat*⁺ and *CRH*⁺ neurons in the POA increase NREM sleep (but not REM sleep) by monosynaptic inhibition of arousal promoting systems that include the TMN, lateral hypothalamus, VTA and DRN.

The authors conclude that this pathway may “provide a possible target for drug development to treat sleep disorders caused by night-time light exposure.”

The study is highly relevant and timely because it advances our understanding of the complex role of the preoptic area in the regulation of sleep and wakefulness. The experiments are well designed, and the data are neat and clearly presented. Interpretations and conclusions are appropriate.

We deeply appreciate the reviewer’s positive comments on the significance of our study.

I have the following comments/questions for the authors’ consideration:

1) In general, it is fair to say that light causes arousal in diurnal species. The role and relevance of the light-induced increase in NREM sleep in mice requires some discussion.

We appreciate the reviewer’s insightful comment. As suggested, we added the following description of the acute light effects on sleep in diurnal animals in the discussion of our revised manuscript: “Nighttime light exposure induces wakefulness in diurnal animals including humans in contrast to sleep in nocturnal species. This highlights neural circuit or molecular differences between diurnal and nocturnal animals downstream of the light input. These differences may be located in the retina and/or the brain. It is intriguing to speculate whether the discovery that ipRGCs that express GABA (Sonoda et al., 2020) may have inhibitory role on POA neurons in diurnal species. For brain mechanisms, the projection patterns of light responsive POA in diurnal animals may differ from those of nocturnal animals. More mechanistic studies are needed to reveal the locus of divergence between diurnal and nocturnal animals in their responses to acute light.”

Do these cells play any role in sleep generation during the daytime?

We appreciate this comment. To examine whether the light-responsive POA neurons play a role in sleep during the daytime, we labeled these neurons with inhibitory DREADDs hM4D(Gi). We then injected CNO in these animals at ZT 2. The CNO injection significantly decreased NREM sleep without influencing REM sleep compared to control (revised manuscript Fig. 3g, h). This shows that light-responsive POA neurons are necessary for NREM sleep during the daytime.

2) Tracing pathways from ipRGC clearly spare the SCN. Did the authors verify -or is there any evidence that suggests- if light-sensitive POA neurons innervate the SCN?

The light-responsive POA neurons do not innervate the SCN as we found no POA fibers or terminals in the SCN (revised manuscript Supplementary Fig. 11c, f).

3) Looks like these cells are contained within a large area of the preoptic region. Are these POA neurons localized within specific anatomical areas/nuclei. Can you be more specific, including an approximate AP range.

We appreciate the reviewer's comment. As shown in the revised manuscript Fig. 2b, the light-responsive POA neurons are located in the medial preoptic area (MPO), the lateral preoptic area (LPO), and the ventrolateral preoptic nucleus (VLPO). The distribution regions of light-responsive POA neurons is similar to that of ipRGC innervations in the POA as shown in our revised manuscript Fig. 1c, d and Supplementary Fig. 1a, b, which is approximately bregma +1.24 mm ~ -0.64 mm. We added this description in our revised manuscript.

4) The authors report dense POA innervation by ipRGC. This is not accurately reflected by Figure 1c. Perhaps they can provide a better image that unequivocally represent the group data.

We appreciate the reviewer's comment. We replaced the original Fig. 1c with revised manuscript Fig. 1 c, d. The reason we use the term 'dense' is that the POA innervations in our study are denser than those shown in published report (Lu et al., 1999). We will change dense to substantial. Importantly, we find that ipRGCs project not only to the ventrolateral preoptic

nucleus (VLPO) but also to the medial preoptic area (MPO) and the lateral preoptic area (LPO) as shown in the revised manuscript Fig. 1c, d.

5) The effect of the activation of light-sensitive POA neurons on NREM sleep is clear, but modest. It would be interesting to compare the quality of NREM sleep by assessing EEG power during NREM sleep between conditions.

As suggested, we now compare the quality of NREM sleep by assessing EEG power during NREM sleep between conditions. We find that the power density of NREM sleep was not affected. This data is shown in the revised manuscript Supplementary Fig. 5.

6) The effect stimulation of night active neurons on REM sleep is very mild (Suppl Figure 3) and too distant from the time of injection. It does not look like a clear drug/stimulation effect, and does not support the interpretation that these cells influence REM sleep.

We agree with this reviewer and as suggested, we deleted the description that activation of night-active POA neurons affect REM sleep.

Additionally, very few neurons were labeled, which raises the question whether it was the scarce number of cells that was not enough to increase NREM sleep. The number of neurons that expressed hM3 receptors in comparison with the other conditions.

This is an important comment that we wanted to address fully. In our revised manuscript Fig. 4 and Supplementary Fig. 10, we find that the cell types of the light-responsive POA neurons are distinct from those of night-active POA neurons. This data show that the low number of cells labeled at night in the POA are consistent with the fact that mice, which are nocturnal, are mostly awake at night. In addition, the distinction between the dark and light-responsive neurons in the POA rule out the possibility that it was the scarce number of cells that cause the lack of NREM sleep effect.

7) Relative to point #2, Lu et al., 2000 (PMID: 10804223) described a subset of preoptic neurons

that expressed Fos during REM sleep. Based on location, they refer to this cluster as “extended VLPO”. Can you tell if light-sensitive POA neurons are present within this area?

We believe that the light-sensitive POA neurons are found beyond the extended VLPO. The data is shown in the revised manuscript Fig. 1c, d and Supplementary Fig. 1a, b. This is also consistent with the ipRGC innervation pattern, which is not only located in the extended VLPO but also in the MPO and the LPO.

8) What was the bin length used for sleep scoring? Please provide this info.

The bin length used for sleep scoring was 10 s. We added this description in the methods of our revised manuscript.

9) It took me a while to see that the SCN and SON in Suppl Figure 2 do not have mCherry expression. Delineating the boundaries of both structures as in Suppl Fig. 4 would make it more evident.

As suggested, we delineated the boundaries of the brain regions in all the figures of our revised manuscript.

10) Was mCherry not enhanced by immunohistochemistry?

mCherry was enhanced by immunohistochemistry. We apologize that the information of this antibody was not included in the original manuscript. We now add this information in the method of our revised manuscript.

11) Please provide the number of mice used in experiments summarized by Figures 3 and 4.

We now add the number of mice used in these experiments in our revised manuscript.

Reviewer #4 (Remarks to the Author):

In this article, Zhang and colleagues report that melanopsin (OPN4) neuronal projections to the preoptic area mediate the sleep-inducing effects of light in nocturnal rodents and that the POA neurons activated by light are CRH⁺ or Tac1⁺. These findings are novel, and they assume great significance as the light is the primary cue that determines the timing and amounts of sleep. Overall, the manuscript is well-written, and the figures are presented clearly. However, I have significant concerns including the sufficiency of datasets for the conclusions derived. In addition, inadequate authentication of methods and the lack of adequate controls largely reduce the enthusiasm. Nevertheless, I believe that the manuscript, when appropriately revised, will be of great interest to the sleep and circadian biology community.

We truly appreciate the reviewer's positive comments on the novelty and significance of our study and the willingness to allow revision. As requested, we now add: 1- Chemogenetic inhibition with the inhibitory DREADDs, hM4D(Gi) of the light-responsive POA neurons and show that it blocks the acute light-induced NREM sleep, but not REM sleep. 2- We show that the light-responsive POA neurons influence NREM sleep during daytime. 3- We show that the night-active POA neurons are completely distinct from the POA neurons that are light-responsive. 4- Using smFISH we show that light-responsive POA neurons do not express galanin and in fact 90% of the light-responsive POA neurons are CRH positive and 10% are TAC1 positive. 5- As mentioned to reviewer 3, we find that ipRGC projections extend beyond the VLPO (ventrolateral preoptic nucleus) to the MPO (medial preoptic area) and the LPO (lateral preoptic area). 6- Finally, we now carry out new control experiment using an AAV virus that expresses GFP but not DREADD. As expected, no NREM changes were observed using this virus to CNO injections. We believe this new data addresses all the concerns raised by this reviewer.

1. Crucial controls are not in place (WT mice controls/AAV controls).

We apologize for this oversight. We now add AAV control (AAVretro-DIO-EGFP) and show the data in the revised manuscript Supplementary Fig. 4. As expected, CNO injection did not affect NREM or REM sleep in these animals.

As for Fig. 2 we have shown the dark control in the supplementary data of the original manuscript and is now shown in the revised manuscript Supplementary Fig. 7.

2. Selectivity and specificity of the AAVs as well as their transduction efficiency was not reported.

We thank the reviewer's helpful reminder. As requested, we now examine the specificity of the AAVs used in Fig. 1 by injecting the same AAVs to wild type (WT) mice. We find that there is no viral expression in the retinas (Supplementary Fig. 2a, b) at all in WT mice, indicating that these AAVs are only expressed in cells that have the Cre recombinase. Moreover, we have shown that the GFP expressing cells in the retina of mice that express Cre in ipRGCs belong mostly to the M1 ipRGC subtype (LeGates et al., 2014). The reconstruction of some of these M1 ipRGCs are shown as new data in revised manuscript Supplementary Fig. 1c. Together this shows that our viral manipulations were specific in labeling the M1 ipRGCs only when Cre is present.

As for the chemogenetic AAVs used in Fig. 2 and revised manuscript Fig. 3, the selectivity and specificity of these AAVs are already provided by the Roth's lab (Krashes et al., 2011).

3. From the figure 1, ipRGC (OPN4)-projections to the POA appear to be restricted to the ventrolateral preoptic region (VLPO); but, do these neurons also project to other POA regions? Figure (s) showing the entire POA and at least 2-3 rostrocaudal levels should be included, which will be helpful to visualize the projection pattern more clearly. If the projections are restricted to the POA, why not use the term 'VLPO'? Apart from POA, where else do these neurons project?

We appreciate the reviewer's insightful comment. To illustrate the ipRGC projections more clearly, we delineate the boundaries of the subregions in the POA and show different rostro-

caudal levels. As shown in revised manuscript Fig. 1c, d and Supplementary Fig. 1a, b, ipRGCs project not only to the ventrolateral preoptic nucleus (VLPO) but also to the medial preoptic area (MPO) and the lateral preoptic area (LPO). After checking ipRGC innervations in the whole brain, we find, not surprisingly, that all the other brain regions known to be innervated by ipRGCs receive projections as has been reported (Ecker et al., 2010). Taken together, ipRGCs innervate the MPO and the LPO, in addition to the VLPO.

How do the authors rule out the involvement of other projections in light-induced sleep?

The way we ruled this out is by limiting our injections to the POA region.

However, we used this comment as a stepping stone to assess whether the POA is also necessary for the acute effects of light on NREM sleep. We labeled light-responsive POA neurons with inhibitory DREADDs hm4D(Gi). We find that when light-responsive POA neurons are chemogenetically inhibited, the acute light pulse cannot increase NREM sleep. This data is shown in the revised manuscript Fig. 3c, d.

These results indicate that the POA is both necessary and sufficient for the acute effects of light on NREM sleep. The necessity and sufficiency of the POA for the acute light strongly argue against the involvement of other regions in the light-induced NREM sleep.

4. Supplementary fig 1 a, b – Although no viral expression is observed in the SCN or SON, significant expression is seen in the subparaventricular zone (SPZ), which also receives retinal projections. Did the authors rule out the contribution of SPZ for the observed effects? Only the cases without significant involvement of SPZ should be included or anatomical controls with injections in the SPZ should be presented.

There is no mCherry expressing cell body in the SPZ as was shown in original manuscript Supplementary Fig. 1 (revised manuscript Supplementary Fig. 3). The fibers that the reviewer implicate are those that arise from the POA neurons transfected by the non-Cre dependent

AAV8-mCherry virus. When we traced our light-responsive POA neurons, we did not observe any innervations in the SPZ (revised manuscript Supplementary Fig. 11).

5. Are there any specific reason why the authors chose ZT14 (rather than ZT12, for e.g.) for injections?

We followed the published conventions. The acute light pulses are usually given from ZT 14 (Altimus et al., 2008; Zhang et al., 2019). To keep it consistent, we also used ZT 14 as the injection time.

In addition, if we give light at ZT12, we will produce different daylength studies (12:12 light dark cycle, changing to 15:9 light dark cycle) and not acute light stimulations.

How were these injections during the dark period performed? If red light was used, the intensity of the light levels may be reported? Even red light above a certain threshold has been shown to modify animal behavior.

We truly appreciate this important point and apologize for the oversight. All injections during the dark period were performed under red light and the intensity of the red light that we used is 8 lx. According to published report, red light dimmer than 10 lx does not influence sleep-wake behaviors in mice (Zhang et al., 2017). We now add this description in the methods of our revised manuscript.

In addition, all control experiments that failed to affect sleep were also performed under the same red light condition.

6. Activation of retinal projections to POA and light pulse-trapped POA neurons promoted SWS. But it is very important to show if inhibition or loss of these neurons prevents the light-induced changes in sleep, without which it is difficult assign a causal role for these neurons in this function.

To answer this important point, we labeled light-responsive POA neurons with inhibitory DREADDs hM4D(Gi). We find that when light-responsive POA neurons are chemogenetically inhibited, acute light pulse cannot significantly increase NREM sleep as shown in revised manuscript Fig. 3c, d.

Similarly, several subsets of neurons in the POA may be sleep-active, which also may be directly or indirectly activated by light. Are the OPN4 neurons specifically target the CRH/Tac1 cells in the POA? Are these neurons selectively mediate the photic effects on sleep or they are part of sleep-circuitry on which light acts to promote sleep?

Although we do not have direct demonstration for this, we believe that the fact that 90% of the light-responsive POA neurons are CRH positive and 10% are TAC1 positive highlights the specificity of the ipRGC input to the POA. In addition, the inhibition of light-responsive POA neurons by inhibitory DREADD showing lack of NREM sleep induction by acute light adds to the conclusion that these neurons are selectively innervated by ipRGCs.

7. As the light exposure increases sleep, light pulse-trapped neurons (cFos-trapped) could just be sleep-active neurons in the POA. As described in the manuscript, sleep in the dark-period is low; thus, activation of fewer POA sleep-active neurons might not be able to induce more sleep than occurs during the dark-period. This raises a concern if the ‘no-light pulse’ controls are sufficient to distinguish ‘sleep-active’ neurons from ‘light-activated’ neurons? This also raises another concern if the ‘night-activated’ neurons were negative for CRH or Tac1? smFish data of night-activated neurons are not presented.

We appreciate the reviewer’s insightful suggestion. As suggested, we characterized the night-active neurons in the POA, and we find that these night-active neurons are not CRH or TAC1 positive (revised manuscript Supplementary Fig. 10). These data indicate that the light-responsive and night-active neurons in the POA belong to distinct cell populations.

8. What percentage of light-responsive neurons in the POA are CRH or Tac1+? As this is not reported, it leads to another question if any of these neurons are galanin+? In addition to CRH,

Tac1 and CCK, galanergic neurons in this region has been associated with sleep for a long time.

We appreciate the reviewer's critical comment. As mentioned above, 90% of the light-responsive POA neurons are CRH positive and 10% are TAC1 positive. However, we did not find any light-responsive POA neurons that are CCK or galanin (GAL) positive as shown in our revised manuscript Supplementary Fig. 9.

9. The telemetry transmitter also provides body temperature data. Did the authors observe any changes in body temperature after activation of either ipRGCs or POA cells? Several populations of POA neurons may alter both sleep and temperature and these variables strongly influence each other.

We appreciate the reviewer's insightful comment. In the original study, we were surprised that light-induced body temperature changes and light-induced sleep did not track similarly with all stimulations. We are convinced that this is an exciting result, on its own merit, and would need to be published once we figure out the brain regions that mediate the effects of light on body temperature.

An example is shown below. Inhibition of the light-responsive POA neurons did not attenuate the acute light effects on body temperature as shown in the figure below (this data will only be included in the rebuttal letter), even though it blunted the effects on NREM sleep (Fig. 3 in the revised manuscript). This indicates that the light-responsive POA neurons are dispensable for the acute light effects on body temperature.

Therefore, according to our results, neural circuits underlying the acute light effects on sleep and body temperature are distinct, indicating that the neural mechanisms underlying the acute regulation of sleep and body temperature by light are much more elaborate than originally suspected (by us and others). Currently, we are trying to define the neural circuits that mediate the acute light effects on body temperature. We hope this reviewer appreciates why it would be premature to present this data before the neural circuit(s) that mediate(s) the acute light effects on body temperature are determined.

References

- Altimus, C.M., Guler, A.D., Villa, K.L., McNeill, D.S., Legates, T.A., and Hattar, S. (2008). Rods-cones and melanopsin detect light and dark to modulate sleep independent of image formation. *Proc Natl Acad Sci U S A* 105, 19998-20003.
- Ecker, J.L., Dumitrescu, O.N., Wong, K.Y., Alam, N.M., Chen, S.K., LeGates, T., Renna, J.M., Prusky, G.T., Berson, D.M., and Hattar, S. (2010). Melanopsin-expressing retinal ganglion-cell photoreceptors: cellular diversity and role in pattern vision. *Neuron* 67, 49-60.
- Krashes, M.J., Koda, S., Ye, C., Rogan, S.C., Adams, A.C., Cusher, D.S., Maratos-Flier, E., Roth, B.L., and Lowell, B.B. (2011). Rapid, reversible activation of AgRP neurons drives feeding behavior in mice. *J Clin Invest* 121, 1424-1428.
- LeGates, T.A., Fernandez, D.C., and Hattar, S. (2014). Light as a central modulator of circadian rhythms, sleep and affect. *Nature Reviews Neuroscience* 15, 443-454.
- Lu, J., Shiromani, P., and Saper, C.B. (1999). Retinal input to the sleep-active ventrolateral preoptic nucleus in the rat. *Neuroscience* 93, 209-214.
- Sherin, J.E., Elmquist, J.K., Torrealba, F., and Saper, C.B. (1998). Innervation of histaminergic tuberomammillary neurons by GABAergic and galaninergic neurons in the ventrolateral preoptic nucleus of the rat. *J Neurosci* 18, 4705-4721.
- Sonoda, T., Li, J.Y., Hayes, N.W., Chan, J.C., Okabe, Y., Belin, S., Nawabi, H., and Schmidt, T.M. (2020). A noncanonical inhibitory circuit dampens behavioral sensitivity to light. *Science* 368, 527-531.

Zhang, Z., Liu, W.Y., Diao, Y.P., Xu, W., Zhong, Y.H., Zhang, J.Y., Lazarus, M., Liu, Y.Y., Qu, W.M., and Huang, Z.L. (2019). Superior Colliculus GABAergic Neurons Are Essential for Acute Dark Induction of Wakefulness in Mice. *Curr Biol* 29, 637-644 e633.

Zhang, Z., Wang, H.J., Wang, D.R., Qu, W.M., and Huang, Z.L. (2017). Red light at intensities above 10lx alters sleep-wake behavior in mice. *Light-Science & Applications* 6.

REVIEWER COMMENTS

Reviewer #1 (Remarks to the Author):

The authors have addressed all the queries raised, including clarifying the time courses used, the amount of sleep induction observed and including new data on cell counts within the POA. The revised manuscript is a novel and extremely valuable addition to this research field.

Reviewer #3 (Remarks to the Author):

All my comments and request were appropriately addressed in the revised version.

Reviewer #4 (Remarks to the Author):

The authors have adequately addressed my previous comments. The authors have included the necessary negative controls, validated the AAVs (but, see below), and provided the additional data requested. The revised manuscript is significantly improved and warrants publication in Nature Communications. I only have a few minor comments.

Chemoinhibition data (Fig 3) presented now are consistent with the chemoactivation results and directly demonstrate a causal role for light-POA neurons in light-induced changes in sleep.

Although the authors claim that they are also 'necessary' for daytime NREM sleep, the effect is minimal (Fig 3g); NREM sleep during the first 3-h combined (similar to Fig 3 d or f) may not be significant. Also, CNO+dark data in Fig 3c-f is absent.

Although authors did not present ipRGC inhibition data, they have presented data from chemoinhibition of light-responsive POA neurons, and it may be sufficient.

The authors mentioned in the rebuttal letter that selectivity and specificity of chemogenetic AAVs were already provided by Roth's lab. Although this is agreeable for Fig 2 and 3, where light-trapped neurons are activated, this is not sufficient for Fig 1. Selectivity and specificity vary across Cre-driver lines, and therefore, quantifying the AAV transduction in ipRGCs is recommended.

Point by point rebuttal to reviewer 4's comments

For clarity, our responses are in blue.

Reviewer #4 (Remarks to the Author):

The authors have adequately addressed my previous comments. The authors have included the necessary negative controls, validated the AAVs (but, see below), and provided the additional data requested. The revised manuscript is significantly improved and warrants publication in Nature Communications.

We appreciate the reviewer's comments.

I only have a few minor comments. Chemoinhibition data (Fig 3) presented now are consistent with the chemoactivation results and directly demonstrate a causal role for light-POA neurons in light-induced changes in sleep. Although the authors claim that they are also 'necessary' for daytime NREM sleep, the effect is minimal (Fig 3g); NREM sleep during the first 3-h combined (similar to Fig 3 d or f) may not be significant.

The effects of daytime inhibition of light-responsive POA neurons on NREM sleep was expected to be small as only one neuronal population of the POA neurons are labeled (CRH-expressing). Since there is a statistical significance between CNO and NS group during ZT 2-3 that does not last beyond one hour, we took away the word necessary for daytime sleep. We changed the title of the section to "Light-responsive POA neurons are necessary for light-induced NREM and influence daytime NREM sleep."

Also, CNO+dark data in Fig 3c-f is absent.

We thank the reviewer for this important experiment, which we now include as Supplementary Figure 9. To determine the role of night-active POA neurons in sleep, we injected 4-OHT in the dark as shown in our revised manuscript Supplementary Fig. 9a. We found that CNO injection did not influence the nighttime light effects on NREM/REM sleep (Supplementary Fig. 9c-f) or

daytime NREM/REM sleep (Supplementary Fig. 9g, h), indicating that the light pulse is required for the labeling of POA neurons that affect NREM sleep.

Although authors did not present ipRGC inhibition data, they have presented data from chemoinhibition of light-responsive POA neurons, and it may be sufficient. The authors mentioned in the rebuttal letter that selectivity and specificity of chemogenetic AAVs were already provided by Roth's lab. Although this is agreeable for Fig 2 and 3, where light-trapped neurons are activated, this is not sufficient for Fig 1. Selectivity and specificity vary across Cre-driver lines, and therefore, quantifying the AAV transduction in ipRGCs is recommended.

We appreciate the reviewer's appreciation of the sufficiency of our chemoinhibition studies.

As for the AAV transduction in ipRGCs, we found that in AAV2-CAG-DIO-GFP experiment, more than 95% ($95.35\% \pm 2.75\%$) of the GFP cells are OPN4 positive (Supplementary Fig. 1c-e), and in AAVretro-Efl α -fDIO-hM3D(Gq)-EGFP experiment, 100% of the EGFP cells are OPN4 positive (Supplementary Fig. 1f-h). This shows that the AAV transduction in ipRGCs is specific and selective. This new data as added as Supplementary Figure 1c-h in our revised manuscript.

REVIEWERS' COMMENTS

Reviewer #4 (Remarks to the Author):

in this revised manuscript, authors have adequately addressed all my previous comments. Addition of histology data and the supplementary figures significantly improved the mansusript. I have no further comments.

I wish to congratulate the authors for their good work!!